# Neonatal and Long-Term Prognosis of Monochorionic Diamniotic Pregnancies Complicated by Selective Growth Restriction

**DOI:** 10.3390/children9050708

**Published:** 2022-05-11

**Authors:** Jessica Mercier, Letizia Gremillet, Antoine Netter, Cécile Chau, Catherine Gire, Barthélémy Tosello

**Affiliations:** 1Department of Neonatal Medicine, North Hospital, Assistance Publique-Hôpitaux de Marseille, 13015 Marseille, France; jessica.mercier@ap-hm.fr (J.M.); catherine.gire@ap-hm.fr (C.G.); 2Department of Gynecology and Obstetrics, North Hospital, Assistance Publique-Hôpitaux de Marseille, 13015 Marseille, France; letizia.gremillet@gmail.com (L.G.); antoine.netter@ap-hm.fr (A.N.); cecile.chau@ap-hm.fr (C.C.); 3CNRS, IRD, IMBE, Aix Marseille Université, 13003 Marseille, France; 4CEReSS, Health Service Research and Quality of Life Center, Aix-Marseille Université, 13005 Marseille, France; 5CNRS, EFS, ADES, Aix Marseille Univ, 13915 Marseille, France

**Keywords:** monochorionic biamniotic twin pregnancies, selective intrauterine growth restriction, neonatal morbidity, neurological outcome

## Abstract

Background: There are few data concerning the neonatal and long-term prognosis of monochorionic biamniotic twin pregnancies (MCBA) complicated by selective intrauterine growth restriction (sIUGR). The aim of the study is to assess the neurological outcomes at two years of age of these newborns and compares these outcomes to those of newborns resulting from intrauterine growth restriction (IUGR) pregnancies. Methods: The study focuses on a cross-sectional prospective cohort of patients treated between 2012 and 2019 in Marseille, France. The primary endpoint is the overall score of the Ages and Stages questionnaires (ASQ) at two years, which assesses the global neurodevelopment. The secondary endpoint is the assessment of neonatal morbi-mortality for both groups (composite endpoint). Results: In total, 251 patients were included in the analysis: 67 in the sIUGR group and 184 in the IUGR group. There was no statistically significant difference in the overall ASQ score at two years but there was the finest motor skills impairment in the IUGR group. The areas most often impaired were communication and fine motor skills. There were no significant differences between the neonatal morbi-mortality of the two groups (adjusted OR = 0.95, *p* = 0.9). Conclusions: Newborns from MCBA pregnancies with sIUGR appear to have similar overall neurological development to IUGR. Notably, IUGR seems to have the most moderate neurobehavioral disorder (fine motor) as a consequence of impaired antenatal brain development due to placenta insufficiency leading to chronic hypoxia.

## 1. Introduction

In total, 10–15% of monochorionic biamniotic twin pregnancies (MCBA) are affected by selective intrauterine growth restriction (sIURG) [1].

The diagnosis of sIUGR has been the subject of a recent consensus: either an estimated fetal weight (EPW) below the third percentile in one of the two fetuses or at least two of the following criteria are met: (1) EFW or (2) abdominal circumference lower than the 10th percentile, (3) a degree of growth discrepancy between the two twins greater than or equal to 25%, and (4) an index of pulsatility of the umbilical artery of the affected twin greater than the 95th percentile [2].

Intrauterine growth restriction (IUGR) occurring in single pregnancies and sIUGR are two different pathologies, but they share similar characteristics, especially regarding their definition. Indeed, in both cases, the diagnosis is based on a fetal growth deemed below the expected weight for gestational age [2,3]. The pathophysiological mechanisms leading to these pathologies are different. The main cause of IUGR is based on a reduced placental blood flow and maternal–placental nutrient and oxygen supply, which result in the fetus’ needs being unmet [4,5]. IUGR involves an abnormal trophoblast invasion, vascular remodeling, reduced placental development and decreased fetoplacental perfusion [6,7,8,9,10]. In contrast, the pathophysiological mechanisms leading to sIUGR are the inequitable division of the placenta and the formation of vascular anastomoses, especially arterio-arterial ones [11,12,13,14]. More often, there is also a velamentous insertion of the cord [15].

The clinical attitudes towards these two pathologies are similar. In both cases, fetal monitoring during pregnancy and birth indications are almost identical, and require regular analysis of fetal growth, heart rate and Doppler abnormalities [1,16,17].

If the consequences of obstetrical prognosis and neonatal and infant morbi-mortality have been well explored for IUGR, there are few studies exploring these areas in sIUGR cases [18].

Associated with sIUGR is an increased risk of in utero death and brain lesions which are linked to the presence of a positive or null umbilical diastole according to the Gratacós’ classification [19,20]. In the literature, data on neonatal outcomes of sIUGR for live-born children are rare, mostly retrospective and limited in number [21]. Similarly, there are few studies on their long-term outcome. These studies most often compare discordant and non-discordant pregnancies, or they compare twins with each other (and not versus single pregnancies with IUGR) [22]. Monitoring these pregnancies means determining the optimal threshold to induce childbirth according to certain ultrasound and clinical criteria [23]. If labor is induced when there are severe Doppler or fetal heart rate abnormalities, there is a risk of fetal hypoxia which adds to the risk of prematurity, thus affecting the newborn’s future [24]. Contrarily, if the delivery is induced before there is any significant fetal damage, there is a risk of prematurity-induced complications.

Having a clearer prognostic differentiation between these two pathologies (IUGR and sIUGR) would make it possible to specify the parental information during the prenatal consultation, to guide obstetrical practices for this complication, as well as to answer questions concerning the newborn’s future.

The main objective of this study is to compare the neurological outcome at two years of age of monochorionic diamniotic twin pregnancies with sIUGR to those IUGR single pregnancy cases, using the French version of the ASQ questionnaire. Our secondary objective was to evaluate the morbi-mortality of newborns in MCBA pregnancies complicated with selective intrauterine growth restriction (sIUGR) and those in single pregnancies with IUGR.

## 2. Materials and Methods

This cross-sectional prospective, analytic, noninterventional study was conducted by the obstetrics-gynecology, maternity, and neonatology departments of the North Hospital in Marseille, Provence-Alpes-Côte d’Azur region (France). Data collection ranged from January 2012 to December 2019 retrospectively and analysis began in January.

### 2.1. Study Population

The two eligible groups for the study were: (i) patients followed at the prenatal diagnostic center for an MCBA pregnancy with sIUGR and (ii) patients with a single pregnancy followed for an IUGR of vascular etiology. All patients were required to be older than 18 years at the time of inclusion. We followed the same protocols for monitoring and birth indications in our department for both groups based on current recommendations. These criteria did not change throughout the study. The follow-up between the two groups was comparable after their birth and was established according to the protocol of the PACA Corsica Monaco perinatal network.

### 2.2. sIUGR Group

sIUGR was defined by Khalil et al., 2019 [2], as EFW of one of the twins below the 3rd percentile, or two or more of the following parameters: (1) EFW or (2) an abdominal circumference lower than the 10th percentile, (3) a degree of growth discrepancy between the two twins greater than or equal to 25%, or (4) if the umbilical artery pulsatility index of the affected twin was greater than the 95th percentile. Regarding the sIUGR group, the criteria for inclusion were: (i) an MCBA twin pregnancy whose monochorionicity, as well as the dating of the beginning of pregnancy, had been confirmed by an ultrasound in the first trimester, (ii) a diagnosis of sIUGR established during pregnancy, and (iii) delivery after 24 weeks of amenorrhea (weeks GA).

The non-inclusion criteria were: fetal death in utero (FDIU) of one or both twins or medical termination of a selective pregnancy.

The exclusion criteria were the occurrence of a twin-to-twin transfusion syndrome (TTTS), twin-anemia polycythemia sequence (TAPS) or a delivery that did not occur at North Hospital.

### 2.3. IUGR Group

The criteria for inclusion were: (1) precise dating by ultrasound in the first trimester, (2) an EFW below the 3rd percentile on adapted curves (curves personalized according to weight, height, age, parity of the mother, fetal gender) or an EFW below the 10th percentile according to these same curves associated with growth arrest or decline [17,25,26], and (3) any delivery after 24 weeks GA. FDIU pregnancies were not included. An infectious, chromosomal, genetic, or malformative cause suspected and/or found during the etiological assessment of growth restriction, and a delivery outside the obstetrics and gynecology department of the North Hospital were also exclusion criteria.

We define “early onset IUGR” (as opposed to “late onset IUGR”) if the age at diagnosis is less than 32 weeks GA [27,28].

### 2.4. Data at Two Years

The exploration of ASQ neurodevelopment is part of our service’s routine care in the follow-up of infants born prematurely and/or with intrauterine growth restriction. An evaluation is programmed at two years of age in a timetable accepted within the PACA (Provence-Alpes-Cote d’Azur), Corsica Monaco perinatal network. This questionnaire, validated in France, was completed by the parents and contains 30 items divided into five evaluation areas: communication, gross motor skills, fine motor skills, problem-solving, and individual and social skills.

A domain is considered impaired if the score obtained in the category is less than −2 SD (standard deviation) from the mean (thresholds noted for each skill), using cut-off points set by the test [29].

Data collection was retrospective and obtained from the child’s medical records for follow-up visits for those children older than two years at the time of inclusion. For children born in 2018, this collection was prospective but performed as part of the routine care already practiced in the department (explained below). These data were also retrieved secondarily from the medical record. There were no additional interventions that modified the child’s care in this study.

### 2.5. Perinatal and Neonatal Data

Maternal, obstetrical and neonatal data were collected retrospectively from medical records following a standardized protocol.

Gestational age was defined as the best estimate from the date of the last menstrual period and/or an ultrasound performed during the first trimester. The other data collected were as follows: maternal age at delivery; gestation and parity; body mass index (BMI), gestational age and growth discrepancy at diagnosis; use of antenatal corticosteroid therapy, complete or not (defined by two injections of corticosteroids 24 h apart); mode of delivery (vaginal or cesarean section); indications for birth according to six main causes (spontaneous or planned, fetal rhythm anomalies (FRA), Doppler abnormalities, growth arrest, pre-eclampsia or another cause).

Growth discrepancy was calculated using the formula: (larger twin EFW–smaller twin EFW ×100)/larger twin EFW. Birth percentiles (weight, height, and head circumference) were calculated from AUDIPOG databases (based on growth restriction curves for weight and height) [30].

Obstetrical care was compliant with the CNGOF recommendations, the main modalities of which were similar for both groups and stayed unchanged [31]. Management in the delivery room followed the recommendations of the International Liaison Committee on Resuscitation (ILCOR) and the French Society of Neonatology (FSN).

Examinations were carried out on a regular basis according to the protocols accepted in the Mediterranean Perinatal Network: transfontanellar ultrasound (TU), electroencephalogram (EEG), cerebral MRI, and ocular fundus (OF).

There are several secondary judgment criteria of interest: (1) comparison of neonatal mortality and morbidity between the two groups according to a composite criterion, (2) neonatal mortality, and (3) neonatal morbidity criteria taken in isolation (Apgar score at 10 min, pH at birth, respiratory distress rate (hyaline membrane disease and bronchopulmonary dysplasia), duration of respiratory support, cerebral abnormality on MRI (ventricular dilatation, intraventricular hemorrhage, periventricular leukomalacia or subependymal hemorrhage), visceral complication (necrotizing enterocolitis [NEC]), sepsis and intensive care unit admission rate).

The composite criterion of interest for neonatal morbi-mortality is the occurrence of one or more of the following events: (1) neonatal death; (2) grade III or IV intraventricular hemorrhage; (3) white matter lesions, a type of cystic periventricular leukomalacia; (4) bronchopulmonary dysplasia defined as the need for oxygen supplementation for at least 28 days, associated with the use of oxygen greater than or equal to 30% and/or ventilatory support (mechanical respiratory assistance or continuous positive airway pressure) at 36 weeks of age-corrected amenorrhea [32]; or (5) ulcerative-necrotizing enterocolitis stage II or III according to the Bell classification.

### 2.6. Statistical Analysis

The sIUGR sample size was limited by the earliest date computer access to our center’s patient records was permitted.

All pregnancies meeting the criteria for sIUGR were followed. The estimated number of twin pregnancies that could be included in this study, given the usual follow-up in our department, was 80. It is difficult to calculate the necessary number of subjects, as we did not find a comparable population in the literature with an evaluation according to the ASQ questionnaire. Moreover, as the study was observational, there were no additional constraints or changes in the management and follow-up of pregnancies and newborns.

For each sIUGR included, at least two “controls” were also included, i.e., two IUGR. Given there was a greater number of IUGR cases than the number of sIUGR cases followed, we stopped recruitment in 2015 to obtain at least twice the number of sIUGR examples. An additional margin of 50% was considered given the estimated loss to follow-up (LFU) at two years.

All tests were bilateral. Results were considered statistically significant if the *p*-value was less than 0.05. This statistical analysis was performed with the 20.0 edition of the SPSS software for the infant data and with the 20.0 edition of the IBM SPSS Statistics software (IBM Inc., New York, NY, USA) for the obstetrical and neonatal data.

Regarding the primary endpoint, the two groups (sIUGR and IUGR) were compared with respect to the overall ASQ score at 2 years of age. Pearson’s chi-squared test was used, and scores from each of the five ASQ domains were assessed.

Subgroups, according to gestational age, were created to compare data at two years for the same gestational age.

According to the newborn’s gestational age, sex, birth weight, and socioeconomic conditions (constructed composite variables: age <25 years and socio-professional category of the mother according to INSEE (Institute of Statistical and Economic Data, Paris, France)), a multivariate analysis was performed.

The two-year analysis began in January 2021, thus children who were not at least two years old at that time were excluded.

Regarding the secondary endpoint, the perinatal data and the composite criterion of morbi-mortality were compared between the two groups (sIUGR and IUGR). Student’s *t*-test (or Mann–Whitney U test if it was necessary) was used for the quantitative variable. Levene’s test was used to check the normal distribution. We used the chi-squared test for categorical variables.

A multivariate analysis was performed in order to compare the neonatal outcome without the differences between the two groups. We made a statistical adjustment on the birth weight, the gestational age at birth, the gender, and indications of birth according to five main causes: spontaneous or planned, fetal rhythm anomalies (FRA), Doppler abnormalities (null intermittent, continuous or positive), growth arrest, or another cause. Results were shown as odds ratios (OR) with a confidence interval of 95% (95% CI).

Since the number of patients lost to follow-up was significant, the perinatal data of the final populations (followed at two years) and the initially included populations were compared with each other to check for attrition bias (Appendix A).

## 3. Results

Initial characteristics of the population184 patients were followed for sIUGR at our center between January 2012 and December 2019. In total, 67 of these were included for neonatal analysis.

There were 264 IUGR patients followed, with 184 being included in the IUGR neonatal analysis group.

Figure 1 shows the flow chart, with patient demographic characteristics as well as the perinatal data presented in Table 1.

The two populations are comparable. The median age at diagnosis was 25–26 weeks GA for sIUGR and IUGR, respectively, with more than 90% of early-onset IUGR in both groups.

There are, however, more severe abnormalities on the umbilical diastole in the sIUGR group: positive in 52.2% of the fetuses of the sIUGR group vs. 73.4% for the IUGR group, null or negative intermittently, respectively, for 17.9% vs. 15.8 %, and null or negative permanently for 29.9% vs. 10.9% (*p* = 0.001).

Neither gestational age nor birth weight differed between the two groups.

The different birth indications are statistically dissimilar between the groups with more induced births for sIUGR (41.8% for IUGRs vs. 16.3% for the spontaneous or planned IUGR births, *p* < 0.001).

### 3.1. The Population’s Evolution at Two Years: Neurodevelopment and Growth

Among the 67 sIUGR evaluated during the neonatal period, 37 (55%) were followed up to the age of two years. Among the 184 IUGR assessed at birth, 75 were followed up to two years of age (41%) (Figure 1). We found the same perinatal characteristics of comparison for those children followed (see additional tables in annex S1a and S1b) as those found for the initial neonatal population (described below).

The median of the global ASQ was not significantly different in either group (250, *p* = 0.84). In the sIUGR group, there were 12 children (32.4%) with an ASQ below the threshold as compared to 18 children in the IUGR group (24%), (*p* = 0.82).

In both sIUGR and IUGR groups, the domains with scores most often below the threshold were communication and fine motor skills. The proportion of children with a score below the threshold decreased as gestational age increased.

In multivariate analysis, there was a significant difference in fine motor skills between the two groups, and they were more significantly pathological in the IUGR group (*p* = 0.02) (Table 2).

Nine sIUGR children (33%) still showed growth retardation below the third percentile at two years as compared to 40 children from the IUGR cohort group (47%) (Figure 2).

### 3.2. Perinatal Characteristics of the Population (Secondary Endpoint)

Appendix A presents the raw neonatal characteristics of the two groups with no significant difference (composite criterion of morbi-mortality: 20.9% for the sIUGR group and 25% for the IUGR group, *p* = 0.5). There was also no statistically significant difference in the neonatal criteria taken separately: pH at birth, mortality rate, resuscitation admission rate, respiratory distress, brain abnormalities, ulcerative colitis, and sepsis (*p* > 0.05).

In the multivariate analysis (Appendix A), we adjusted different parameters: birth weight, gestational age at birth, gender, indication of birth (divided into five categories: spontaneous or induced delivery, fetal heart rate abnormalities, fetal Doppler abnormalities, fetal growth arrest, and other causes) and the last umbilical diastole before birth (intermittent or permanent positive, null or negative).

There is a difference in mortality (9% in the sIUGR group versus 6% in the IUGR group), but it is not statistically significant (*p* = 0.06). In all results, there is no significant difference, even with morbidity and the composite criteria of morbi-mortality: *p* = 0.92 (Appendix A) in the multivariate regression model.

### 3.3. Eutrophic Twin Data

Table 3 compares the two-year outcomes of the small twin with that of the eutrophic twin.

There is no significant difference except for a trend in the number of altered domains (*p* = 0.07) with a tendency for communication impairment (0.07) for the sIUGR group.

Appendix A) shows the perinatal outcome of the eutrophic twin.

## 4. Discussion

This retrospective study compares the perinatal and long-term outcomes (2 years) of children followed either for sIUGR in MCBA twin pregnancies or for IUGR in a single pregnancy.

The results of our study show that the long-term neurobehavioral evolution at 2 years on the overall ASQ is similar between the sIUGR group and the IUGR group, except for the more pathological fine motor domain in the IUGR group. The two most impaired ASQ domains in the total population are in relation to communication (13%) and fine motor skills (8%).

Neonatal morbidity is similar between the two groups; the sIUGR group seems to have a difference in mortality but it is not statistically significant.

### 4.1. Study Strengths and Limitations

In the literature, there are further studies concerning IUGR neurobehavioral outcomes. Based on those studies, we can hypothesize that growth discrepancy in monochorionic pregnancies is a separate clinical entity whose consequences begin in the antenatal period, which is reinforced by Gratacos’ Doppler studies [19]. The use of a standardized two-year evaluation questionnaire, validated by the American Academy of Pediatrics (AAP), in France and used in a French national prospective study (Epipage 2) has enabled us to make reliable developmental comparisons [34,35,36,37]. There are few studies evaluating the neonatal and long-term outcomes of neonates born from twin pregnancies with sIUGR and, to our knowledge, this study is the first comparing these children with those born as singletons with vascular IUGR. We performed a multivariate analysis taking into account potentially confounding factors known to influence neonatal prognosis, such as gestational age, sex, weight, and Doppler abnormalities. This allows us to increase the reliability of the comparison. In this statistical adjustment, we also considered the socio-economic status of the mother which can have both positive and/or negative impacts on children’s long-term futures.

The retrospective nature of our study, the small number of children included in the sIUGR group and the high ratio of those lost to follow-up (especially in the IUGR group) are limitations to the interpretation of our study. We assess infants until 24 months on their neurodevelopment, but we have no data on the longer-term prognosis during childhood (it would be interesting to continue the follow-up).

We did not define a gestational age limit for inclusion, and therefore included both very preterm and full-term newborns in order to obtain a sufficient number of data (and therefore sufficient statistical power). Pregnancies with in utero death were excluded from the analysis to focus on the neonatal and long-term prognosis of the sIUGR twin. Therefore, the most severe clinical situations were not assessed in the analysis [38]. We made this methodological choice in order to specifically focus our study on neonatal morbidity and mortality and the neurological outcome of the smallest twin with sIUGR.

### 4.2. Literature

A review of the recent literature in 2019 [39], whose objective was to assess the impact of sIUGR on the long-term neurological development of twins resulting from monochorionic pregnancies (MC), mainly shows the current lack of knowledge on the subject. Of the 28 articles assessed for eligibility, only five were included. One article [40] showed that the incidence of long-term neurodevelopmental impairment (NDI) is higher at 2 years in discordant MCBA (11/26, 42%) versus dichorial (DC) (5/38, 13%) and concordant MCBA (6/71, 8%), *p* < 0.01. MCBA discordant twins had a six times higher risk of cerebral palsy as compared to DC twins (5/26, 19% versus 1/40, 3%, *p* < 0.05). Note that in our work, we do not study cerebral palsy because of the relatively high gestational age of our population, but we have a greater tendency to death, signifying great antenatal severity. The finest motor impairment in the IUGR population is consistent because fine motor impairment is predictive of a more globally associated neurobehavioral impairment at school age, as described in the IUGR population [41]. In addition, a fine motor disorder serves as a good predictor of later school difficulties, as cognitive faculties and fine motor skills are intimately linked [42] and involve cerebral areas co-activated in the performance of numerous activities [43]. Indeed, children with IUGR have more neurobehavioral disorders, including attention disorders and learning disabilities [34,44]. These apparently healthy children show difficulties with attention, behavior and interpersonal relationships [41]. Edmonds et al. [45] describe a linear relationship between verbal IQ scores and birth weight, showing a 13-point difference for a 1000 g growth discrepancy between twins, at the expense of the smaller twin (*p* < 0.0001). The verbal IQ score is lower in the smaller twin versus the larger twin (0.5 SD less). There is the same trend, but not significantly, for ASQ in the communication domain in our study. Sierakowski et al. [46] show that MCBA twins without prenatal complications would be ultimately at risk of subtle neurodevelopmental difficulties that are associated with a lower birth weight. An increased impairment in language and neurobehavioral disorders could depend on a low gestational weight for the age rather than a weight discrepancy. Finally, in three articles analyzing intra-pair differences, it is the smaller twin who more frequently presented with mild NDI (14) (6/80, 8% versus 1/111, 1%, *p* = 0.02), lower developmental test scores (up to 2.2 points) [23] and lower reasoning skills (up to 7 points) [47] compared to the larger twin.

All these results suggest that monochorionic (MC) twins with sIUGR have a higher risk of long-term neurodevelopmental impairment as compared to concordant dichorionic or monochorionic twins, and the smaller twin seems to be more affected than his co-eutrophic twin. Despite all these data, the overall level of evidence seems insufficient and needs further exploration.

Boghossian et al. [44] evaluated twin pregnancies with sIUGR at 18–22 months. The smaller twin presented with increased mortality, risk of developmental delay, lower weight and a smaller head circumference than twins resulting from pregnancies with concordant weight, and these results are similar to those of our study.

However, other studies find different long-term outcome results. Monset-Couchard et al. [48] evaluated 36 newborns from twin pregnancies with sIUGR at three years and found no difference in locomotor scores; however, they had more language disorders. C.Vedel et al., in a single large observational study concerning 1134 twins [22], seemed to show an increase in mortality in the context of sIUGR but without any difference in neurological evolution on the long-term global ASQ score. It should be noted that in this study, no analysis is made in the ASQ sub-domains and, identically, our study does not show a difference in the global ASQ score between the sIUGR and IUGR singletons. Nevertheless, in Vedel’s study, neurodevelopmental disorders seemed more related to hypotrophy and not to growth discrepancy, thus implying chorionicity in their sub-analysis.

However, in the literature, there is no study, to our knowledge, comparing MCBA twin pregnancies with sIUGR and single pregnancies with vascular IUGR. If we assume that the IUGR pregnancies have more frequent neurodevelopmental problems centered on attentional abilities and academic performance, our results confirm greater fine motor impairment in IUGR pregnancies and suggest that the growth discrepancy in monochorionic pregnancies appears to play a role antenatally in death, but hypotrophy correlates to neurodevelopment at two years of age [49,50,51]. As a reminder, Rustico et al. [14] followed 80 newborns and found more minor deficits in language or motor skills in the sIUGR twin (8%) as compared to the co-twin (1%). In the Epipage 2 [34] cohort, a study has been performed on the lexical stock of preterm infants. They note a strong association between the language skills and abilities in other developmental domains when the children reach 24 months of age. This finding confirms that each domain cannot be considered only separately: impairment in one skill can impact the development of the child’s other abilities, and neurodevelopment needs to be assessed from a holistic approach during infancy [52].

Pre-linguistic sensorimotor skills are necessary for the child’s language development. Impairment of sensorimotor functions (disorders of orofacial praxis, alteration of auditory discrimination, sense of touch, visual attention) is noted in children with language disorders [53]. Our findings of fine motor impairment in the IUGR group must be interpreted in a more global neurodevelopmental way.

All in all, few studies examine sIUGR in monochorionic pregnancies. Those assessments are heterogeneous, with limited numbers, and the comparison populations are varied between co-twin, dichorionic pregnancy, and concordant MC pregnancy. However, our study seems to confirm the findings of other studies indicating the tendency for a more adverse outcome for the discordant twin, favored by monochorionicity. This is confirmed by a recent study finding a higher risk of adverse outcomes in MC twins but no differences in the neurodevelopmental follow-up at 2 years between MC and DC twins [38].

Thus, one hypothesis suggests that neurobehavioral disorders in premature IUGR infants are due to cerebral hypoconnectivity secondary to an encephalopathy of prematurity. This begins prenatally because of placental vascular disturbances. Monochorionicity with sIUGR could lead to more deaths and/or to more diffuse disturbances and cerebral structural abnormalities at an early stage [54].

MCBA twin pregnancies with sIUGR require regular monitoring, including umbilical diastole Doppler assessment, to assess the risk of a poor prognosis and to estimate the optimal threshold for an induced delivery [14,21]. To ensure a more effective intervention, it is important to identify these children early on and to follow them throughout their development [55,56]. It would be interesting to confirm these results with a larger cohort study.

## Figures and Tables

**Figure 1 children-09-00708-f001:**
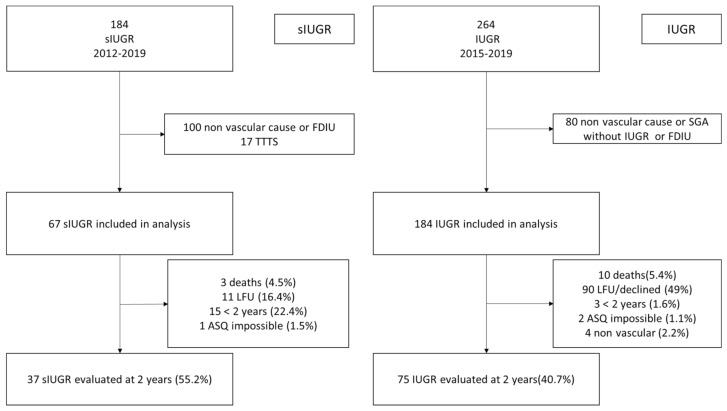
Flow chart. IUGR: intrauterine growth restriction. sIUGR: selective intrauterine growth restriction. FDIU: fetal death in utero. SGA: small for gestational age. LFU: lost to follow-up, TTTS: twin to twin transfusion syndrome. ASQ: ages and stages questionnaires. Impossible: hemophilia complicated by stroke or autism spectrum disorders. Four IUGR excluded secondarily: secondary diagnosis chromosomal or genetic abnormality.

**Figure 2 children-09-00708-f002:**
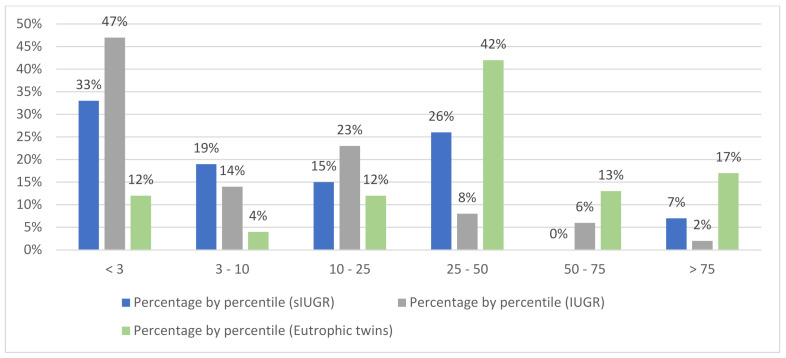
Weight percentile distribution at 2 years of age. sIUGR: selective intrauterine growth restriction. IUGR: intrauterine growth restriction (single pregnancy). Percentile defined from WHO (World Health Organization) growth charts for age 2 years, and according to gender data on 27 sIUGR, 86 IUGR, 24 eutrophic twins.

**Table 1 children-09-00708-t001:** Group comparisons of demographic and pregnancy characteristics between the “diamniotic monochorionic twin pregnancy with selective intrauterine growth restriction” and “single pregnancy with intrauterine growth restriction”.

	sIUGR	IUGR	*p*-Value
**Number of patients, *n***	67	184	
Mother’s age, years, mean ± SD	29.7 ± 5.4	30.0 ± 6.2	0.67
Gravidity, *n*, median (min–max)	2 (1–10)	2 (1–11)	0.90
Parity, *n*, median (min–max)	0 (0–5)	0 (0–10)	0.81
BMI, kg/m^2^, mean ± SD	24.5 ± 5.4	26 ± 6.1	0.08
Gestational age at diagnostic, Weeks of amenorrhea, median (min–max)	25 (12–37)	26 (18–36)	0.08
Growth discrepancy at diagnosis, %, mean ± SD	22.4 ± 7	-	
Growth percentile at diagnosis, %, mean ± SD	-	4.3 ± 5.1	
Gender, *n* (%)			0.28
Male	39 (58.2%)	93 (50.5%)	
Female	28 (41.8%)	91 (49.5%)	
Antenatal corticosteroid, *n* (%)			0.38
Not carried out	23 (34.3%)	69 (37.5%)	
Incomplete	1 (1.5%)	9 (4.9%)	
Complete	43 (64.2%)	106 (57.6%)	
Delivery, *n* (%)			0.41
Vaginal	21 (31.3%)	48 (26.1%)	
Cesarean	46 (68.7%)	136 (73.9%)	
Birth indications, *n* (%)			<0.001
Spontaneous birth or scheduled cesarean	28 (41.8%)	30 (16.3%)	
Cardiac fetal anomaly	18 (26.9%)	65 (35.3%)	
Doppler anomalies	9 (13.4%)	5 (2.7%)	
Growth arrest	9 (13.4%)	65 (35.3%)	
Pre-eclampsia	2 (3.0%)	19 (10.3%)	
Other	1 (1.5%)	0 (0.0%)	
Gestational age at birth, Weeks of amenorrhea, median (min–max)	34 (25–38)	33 (26–42)	0.98
Last umbilical diastole before birth, *n* (%)			0.001
Positive	35 (52.2%)	135 (73.4%)	
Null intermittent	12 (17.9%)	29 (15.8%)	
Null permanent	20 (29.9%)	20 (10.9%)	
Birth weight, grams, mean ± SD	1545 ± 567	1475 ± 670	0.45
Growth discrepancy at birth, %, mean ± SD	22.6 ± 12.0	-	-
Growth percentile at birth, %, mean ± SD	2.2 ± 2.4	1.7 ± 2.2	0.16
Number of newborns	35	130	
Percentile of height at birth, %, mean ± SD	10.9 ± 10	7.4 ± 15.3	0.24
Percentile of HC at birth, %, mean ± SD	14.7 ± 19.1	8.8 ± 13.5	0.33

sIUGR: selective growth restriction. IUGR: intrauterine growth restriction (single pregnancy). HC: head circumference. SD: standard deviation. BMI: body mass index. *p*-value considered statistically significant if <0.05. Table from L. Gremillet et al. [33].

**Table 2 children-09-00708-t002:** Comparison of 2-year ASQ scores between sIUGR and IUGR, overall and by gestational age subgroups.

	Global			26–31 Weeks GA			32–34 Weeks GA			≥35 Weeks GA		
	sIUGR, *n* = 37	IUGR, *n* = 75	* *p*-Value	sIUGR, *n* = 11	IUGR, *n* = 29	* *p*-Value	sIUGR, *n* = 14	IUGR, *n* = 15	* *p*-Value	sIUGR, *n* = 12	IUGR, *n* = 31	* *p*-Value
**ASQ**												
Median, [IQR]	250 [230–260]	250 [235–265]	0.84	235 [225–247.5]	245 [230–255]	0.31	240 [230–260]	250 [220–260]	0.52	257,5 [250–265]	250 [235–270]	0.39
ASQ ≤ 220, *n* (%)	9 (24.3)	13 (17.3)	0.92	2 (18.2)	7 (24.1)	0.56	3 (21.4)	4 (26.7)	0.75	1 (8)	2 (6.5)	0.69
ASQ < threshold, *n* (%)	12 (32.4)	18 (24)	0.82	7 (63.6)	9 (31)	0.43	4 (28.6)	4 (26.7)	0.37	1 (8)	5 (16.1)	0.49
Number of altered domains, *n* (%)												
0	28 (75.7)	58 (77.3)	0.45	4 (36.4)	20 (69)	0.33	12 (85.7)	12 (80)	0.55	12 (100)	26 (83.9)	0.33
1	8 (21.6)	12 (16)		6 (54.5)	5 (17.2)		2 (14.3)	3 (20)		0	4 (12.9)	
2	1 (2.7)	4 (5.3)		1 (9.1)	3 (10.3)		0	0		0	1 (3.2)	
3	0	0		0	0		0	0		0	0	
4 or 5	0	1 (1.3)		0	1 (3.4)		0	0		0	0	
At least one altered domain, *n* (%)	9 (24.3)	17 (22.7)	0.76	7 (63.6)	9 (31)	0.098	2 (14.3)	3 (20)	0.79	0	5 (16.1)	0.57
By domain, *n* (%)												
Communication	5 (13.5)	7 (9.3)	0.39	3 (27.3)	4 (13.8)	0.31	2 (14.3)	1 (6.7)	0.28	0	2 (6.5)	0.68
Gross motor skills	1 (2.7)	4 (5.3)	0.67	1 (9.1)	3 (10.3)	0.98	0	0	0.45	0	1 (3.2)	0.44
Fine motor skills	3 (8.1)	8 (10.7)	0.02	3 (27.3)	5 (17.2)	0.16	0	1 (6.7)	0.069	0	2 (6.5)	0.38
Problem solving	0	2 (2.7)	0.65	0	1 (3.4)	0.64	0	1 (6.7)	0.50	0	0	0
Social and individual aptitudes	1 (2.7)	3 (4)	0.89	1 (9.1)	2 (6.9)	0.68	0	0	0.41	0	1 (3.2)	0.84

sIUGR: selective intrauterine growth restriction. IUGR: intrauterine growth restriction (single pregnancy). *p*-value considered statistically significant if <0.05. ASQ: ages and stages questionnaire, validated by the American Academy of Pediatrics, pathological if below threshold (either ASQ ≤ 220 or at least one domain impaired), a domain being considered impaired if <2 standard deviations from the mean. IQR: interquartile range. * Multivariate analysis with adjustment for gestational age, birth weight, fetal gender, and socioeconomic status.

**Table 3 children-09-00708-t003:** Comparison of eutrophic twins with sIUGR co-twins at 2 years of age (global ASQ).

ASQ	Eutrophic (N = 36)	sIUGR (N = 37)	*p*-Value
**Median**	255	250	0.37
IQR small	243.75	230	
IQR large	271.25	260	
**ASQ ≤ 220**	3 (8.3)	9 (24.3)	0.48
**ASQ < threshold, *n* (%)**	5 (13.9)	12 (32.4)	0.26
**Number of altered domains** **, *n* (%)**			
0	33 (91.7)	28(75.7)	0.11
1	2 (5.6)	8 (21.6)	
2	0	1 (2.7)	
3	1 (2.8)	0	
4 or 5	0	0	
**At least one altered domain** **, *n* (%)**	3 (8.3)	9 (24.3)	0.07
**By domain** **, *n* (%)**			
Communication	0	5 (13.5)	0.05
Gross motor skills	1 (2.8)	1 (2.7)	0.13
Fine motor skills	2 (5.6)	3 (8.1)	0.18
Problem solving	1 (2,8)	0	0.23
Social and individual aptitudes	1 (2.8)	1 (2.7)	0.96

N, *n*: number. sIUGR: selective intrauterine growth restriction. IUGR: intrauterine growth restriction (single pregnancy). IQR: interquartile range. *P*-value considered statistically significant if < 0.05. ASQ: ages and stages questionnaire, validated by the American Academy of Pediatrics, pathological if below threshold (either ASQ ≤ 220 or at least one domain impaired), a domain being considered impaired if < 2 standard deviations from the mean.

## Data Availability

The datasets that generated and/or analyzed during the current study are not publicly available due to the data belongs to the Assistance Publique Hopitaux de Marseille. However, datasets are available from the sponsor (promotion.interne@ap-hm.fr) on reasonable request and after sign a contract pertaining to the provision of data and/or results. More information on gyneco-obstetrical and perinatal data is available in the article by L. Gremillet et al. [33].

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
