# Peer review of "Neonatal and Long-Term Prognosis of Monochorionic Diamniotic Pregnancies Complicated by Selective Growth Restriction"

_children, 2022, doi:10.3390/children9050708_

Round 1

Reviewer 1 Report

This is a well written article concerning the neonatal and long-term prognosis of monochorionic diamniotic twin pregnancies complicated by selective intrauterine growth restriction. As a mean of comparison the prognosis of singleton FGR fetuses is used. The study design and statistical analysis is adequate. There are tables summarizing the findings. The conclusions are supported by the results. limitations of the study are included. No spelling issues apart from IURG that should be IUGR or even better FGR. I think the paper adds significant information in counselling.

Author Response

Dear Doctor, Dear Reviewer,

We thank you so much for your valuable and appreciated inputs in our article. The interest you showed in this regard encourages us to continue our research.

The acronym IUGR has been spelled correctly.

Respectfully

Reviewer 2 Report

I am pleased to have the possibility to review the study “Neonatal and long-term prognosis of monochorionic diamniotic pregnancies complicated by selective growth restriction”. The discussed problem of low birth weight is crucial, has a meaningful impact on maternal and children’s health, and requires further analysis. The size of the study population, time range (7 years) of included patients and the prospective character of the study are undoubtful study strengths. Nevertheless, the methodological issues of the study do not allow to approve of the study in the current form for publication.

The abstract should be paraphrased and rewritten. Firstly, the aim of the study is not clearly present. It does not show what groups were compared. The study's objective was also not answered correctly in the result section. The primary endpoint was called “...the overall score of the Age and Stages Questionnaire (ASQ) at two years…” - It should be more specified what the final score of ASQ means and called, e.g. as neurodevelopment of the children.

What do you mean with the expression “…trend to excess death…”? I haven’t seen it in your results.

Objectives should be presented as the last paragraph of the Background, not as part of the methodology.

The definition used in the study is outdated because, according to new recommendations, the IUGR spectrum should be defined according to new recommendations as SGA, early-onset FGR and late-onset FGR. Please standardise your manuscript to the latest recommendations[1–3].  You cite Bennasar M al. published in Seminars in Fetal & Neonatal Medicine, 2017, which do not seem to be a systematic revive or recommendations that are strongly recommended by definition showing.

A critical bias of the study is its methodology. Comparison of early-onset FGR and late-onset FGR seems to be a considerable limitation of the study. I suppose to use a more appropriate comparison group should be made. Early-onset FGR Geminis could be compared to late-onset Geminis or early-onset Geminis to early-onset Singleton pregnancies. It seems to be a solid methodological issue of the study, which needs reconstruction of the whole paper.

You used the ASQ scale. Has it have been evaluated for the French society? Please add an appropriate citation if it was so.

Morbidity and mortality, in my opinion, cannot be assessed as one outcome. mortality is more crucial as an endpoint according to its consequences. Neurodevelopment is more seems to be a secondary outcome.

 The deep concern is why the data was collected respectfully in a prospective study? It is unclear and needs clarification.

A detailed description of TTS, TRAP etc., is not needed in my opinion when you exclude it from the study.

Perinatal data 185-195 is also part of the statistic. I would suggest the resignation of subtitles in this place.

It was mentioned in the methodology that the multivariate regression model was built. Nevertheless, the results of this mode were not presented in the study. Please clear this.

Major points:

  1. Comparing early-onset FGR in Geminis and late-onset FGR in singleton pregnancies seems inappropriate.
  2. Actualisation and standardisation definition of fetal growth retardation according to latest recommendations.
  3. Primary and secondary outcomes should be reconsidered.

Minor points:

  1. The abstract should be paraphrased and rewritten because of the aim of the study and methodology description.
  2. Correction of the study structure is needed.
  3. Has an ASQ have been evaluated for the French society?
  4. Why were the data collected respectfully in a prospective study?
  5. Lack of multivariate regression model description.

Reviewer 3 Report

Excellent article with a very good design. The authors have managed to show that selective IUGRs may have increased mortality vs IUGRS in monochorionic diamniotic pregnancies . These two different modalities have different pathophysiology. I congratulate the authors for the hard work. 

Author Response

Dear Doctor, Dear Reviewer,

We thank you so much for your valuable and appreciated inputs in our article. The interest you showed in this regard encourages us to continue our research.

Respectfully

Round 2

Reviewer 2 Report

Thank you for your careful editing of the manuscypt. Nevertheless, there are some issues that needs further clarifications: 

  1. Thank you for your structure amendments in the abstract. Nevertheless, it remains still confusing that the study's secondary endpoint, even insignificant, is more crucial for maternal-neonatal health than a primary endpoint. According to the literature, “Secondary outcomes are particularly helpful if they lend supporting evidence for the primary endpoint.

Vetter, Thomas R., and Edward J. Mascha. 2017. ‘Defining the Primary Outcomes and Justifying Secondary Outcomes of a Study: Usually, the Fewer, the Better. Anesthesia and Analgesia 125 (2): 678–81. https://doi.org/10.1213/ANE.0000000000002224.

Please, correct me if I misunderstood the endpoint of your study and its objectives.

  1. a. I understand your point about the difference in definitions between FGR and IUGR.

The latest recommendations about growth restrictions were not added to my previous review:

Figueras, Francesc, and Eduard Gratacós. 2014. ‘Update on the Diagnosis and Classification of Fetal Growth Restriction and Proposal of a Stage-Based Management Protocol’. Fetal Diagnosis and Therapy 36 (2): 86–98. https://doi.org/10.1159/000357592.

Lees, C. C., T. Stampalija, A. Baschat, F. da Silva Costa, E. Ferrazzi, F. Figueras, K. Hecher, et al. 2020. ‘ISUOG Practice Guidelines: Diagnosis and Management of Small-for-Gestational-Age Fetus and Fetal Growth Restriction. Ultrasound in Obstetrics & Gynecology: The Official Journal of the International Society of Ultrasound in Obstetrics and Gynecology56 (2): 298–312. https://doi.org/10.1002/uog.22134.

Vayssière, C., L. Sentilhes, A. Ego, C. Bernard, D. Cambourieu, C. Flamant, G. Gascoin, et al. 2015. ‘Fetal Growth Restriction and Intra-Uterine Growth Restriction: Guidelines for Clinical Practice from the French College of Gynaecologists and Obstetricians’. European Journal of Obstetrics, Gynecology, and Reproductive Biology 193 (October): 10–18. https://doi.org/10.1016/j.ejogrb.2015.06.021.

  1. b. As I mentioned above, this difference is not so crucial for interpreting the results. But I fill like my comment, “Early-onset FGR Geminis could be compared to late-onset Geminis or early-onset Geminis to early-onset Singleton pregnancies. It seems to be a solid methodological issue of the study, which needs reconstruction of the whole paper.”

I assume it’s the most crucial issue of the manuscript. Because the two groups compared in the study were different in two different aspects. It was not only because of the comparison “sIUGR vs IUGR” but also as “twins vs singleton pregnancies”. I think it should be firstly clarified before resubmission. As you mentioned in discussion sections 422-423. There is a reason not to compare two different groups.

  1. I assume the main objective's definition and additional analysis were combined with the primary and secondary outcomes.
  2. Thank you, now I understand
  3. It should be mentioned in the exclusion criteria of the study.
  4. Thank you. I suppose this opinion that “no significant differences in the multivariate regression model were found should be placed in the main text”. No statistical differences in the multivariate regression model show us that results, which were significant in the simple log regression model, were correlated with ich other. Could you clarify what the aim of the multivariate regression model performance was?
  5. I strongly suggest following the STROBE guidelines for the cohort studies in your study. Firstly, in the discussion section.
  6. The discussion sections do not aim to repeat the methodological section of the study. The main result should be shortly presented than discussed in comparison to what is known with presenting limitation of the study, as it is clearly described in the STROBE protocol (https://www.equator-network.org).
